# Efficiency of Diagnostic Test for SARS-CoV-2 in a Nursing Home

**DOI:** 10.3390/geriatrics7040078

**Published:** 2022-07-26

**Authors:** Sergio Salmerón, Alonso López-Escribano, Inmaculada García-Nogueras, Joaquina Lorenzo, Juan Manuel Romero, Antonio Hernández-Martínez, Francisco García-Alcaraz

**Affiliations:** 1San Vicente de Paúl Nursing Home, Diputación de Albacete, 02001 Albacete, Spain; i.garcia@dipualba.es (I.G.-N.); j.lorenzo@dipualba.es (J.L.); jm.romero.sv@dipualba.es (J.M.R.); 2Centro de Salud Bonete, Gerencia de Atención Integrada de Almansa, Servicio de Salud de Castilla-La Mancha, 02691 Albacete, Spain; alonsol@sescam.jccm.es; 3Faculty of Nursing of Ciudad Real, Universidad de Castilla-La Mancha, 13003 Ciudad Real, Spain; antonio.hernandez@uclm.es; 4Nursing School, Universidad de Castilla-La Mancha, 02071 Albacete, Spain; fgarciaalcaraz@gmail.com

**Keywords:** coronavirus, COVID-19, nursing homes, cost-efficiency, SARS-CoV-2

## Abstract

Background: there is no consensus on how to optimally use diagnostic tests in each stage of COVID-19 pandemic. The objective of this research is to determine the efficiency of sorting positive antibody test quarterly. Methods: this research uses a retrospective, observational study. COVID-19 diagnostic tests performed and avoided refer to a Spanish nursing home. Population: 261 employees and 107 residents. A quarterly antibody test was performed on subjects who had tested positive during the first wave of coronavirus, and a antibody rapid test on the remaining subjects. Results: during the first wave, 24.0% of the employees and 51.4% of the residents had a positive antibody test. Seronegativization was observed in 7.6% of employees and 1.6% of residents. An employee was infected with COVID-19 in September 2020, followed by a nursing home outbreak in October: 118 Polymerase Chain Reactions tests were avoided in residents and 18 in employees, which in turn prevented 15 workers from going on sick leave and the quarantine of 59 residents. This represents savings of about $15,000. Conclusions: our study supports the need to know and apply the strategies for early detection, surveillance and control of COVID-19 for future outbreaks. We conclude that surveillance for positive COVID-19 serology among long-term care staff and residents may be a cost-effective strategy during a pandemic.

## 1. Introduction

On 7 January 2020, Chinese authorities identified a new type of coronavirus called SARS-CoV-2 [1], the World Health Organization (WHO) named this new disease COVID-19, and in March 2020 WHO declared its outbreak a pandemic [2]. The impact of COVID-19 has been most significant on the elderly residents at nursing homes [3,4]. This population has also suffered more severe clinical, functional and psychological complications [5,6] than the general population. In Spain, once the first wave of the pandemic (between March and June 2020) was overcome, the overall national prevalence was around 5% [7]. In addition, the proportion of people with immunoglobulins G (IgG) against SARS-CoV2 was higher in elderly residents of large cities (>100,000 inhabitants), estimated at 6.8%. After the first wave, nursing homes had the following tests for coronavirus diagnosis:-Polymerase Chain Reaction (PCR): detects the presence of coronavirus-specific genetic material in oropharyngeal and nasal samples, with a specificity of 99% [8].-Rapid immunochromatography test (RT): This is an antibody test, it allows obtaining results in situ and does not require venipuncture. Based on reliability studies, RT exhibits a sensitivity of 69.6% for IgM and 82.1% for IgG, with a specificity of 99% for IgM and 100% for IgG [9].-IgG antibody serology: performed by microparticle chemiluminescent immunoassay, which is more sensitive than RT but requires venipuncture. Sensitivity can reach 100% in confirmed cases after 14 days from the onset of symptoms, with a specificity of 99.6% [10].

The Spanish authorities recommend carrying out diagnostic tests prioritizing health, sociohealth and educational centers [11] as well as prioritizing vulnerable patients. Thus, in nursing homes, in an outbreak it may be advised to obtain PCR [12] from all residents and employees. After the first wave, on 1 June 2020, the Spanish Ministry of Health indicated that people with symptoms compatible with COVID-19 who have already had an infection confirmed by active coronavirus infection detection test (AIDT: PCR or RT of antigens) or IgG serology of positive SARS-CoV-2 in the previous 90 days, would not be considered suspected cases again, except if they exhibited COVID-19 symptoms of high suspicion [11] (Figure 1). This strategy changed on 18 December 2020 [13] where authorities recommended not to perform AIDT on patients with positive AIDT in the previous 90 days. However, AIDT would still be carried out even if an individual had a previous positive IgG antibody serology. Given that requesting an AIDT was independent of whether or not IgG antibody serology was positive, there was no reason to keep recording the positive IgG antibody serology quarterly. This change is probably due to the concern that premature reinfections may appear due to new strains of SARS-CoV-2 [14].

Many studies have considered what would be the most efficient strategy for the use of diagnostic tests for coronavirus according to the different stages of evolution of the pandemic. Some have addressed the advantages and limitations of each diagnostic method [15], but there are no studies focused on residential environments nor has a consensus been reached on the issue [16]. This study is necessary to help us design future diagnosis and screening strategies.

The principal objective is to determine the efficiency of the strategy of serialization of positive serologies for coronavirus on a quarterly basis in order to avoid performing unnecessary AIDT, sick leave and quarantines. Secondary objective: To evaluate, in employees, if the COVID-19 infection (PCR or positive IgG serology) is related to age, sex or job position.

## 2. Materials and Methods

Population and sample. Retrospective observational study. Sample: Employers or institutionalized subjects who are working or residing in the San Vicente de Paúl nursing home (Albacete, Spain). We describe the complementary tests, carried out and avoided, for the diagnosis of COVID-19 of a selected sample of 107 residents and 261 employees, from June 2020 to 18, December 2020. Sources of information: the medical clinical history was selected. This contains all the results of complementary tests, quarantines and sick leave. Inclusion criteria: Being a resident or employee of the social health center and signing an informed consent. Exclusion criteria: Refusal to participating in the study and/or signing of the informed consent.

Variables. They were collected in order to describe the general characteristics of the selected population: age, sex, pathologies of residents and job position of employees (healthcare and non-healthcare). In order to analyze the cost-efficiency of the study and following the protocols of the Spanish Ministry of Health, the results and number of coronavirus diagnostic tests performed (PCR, serology and RT) were collected, as well as the PCR, RT, quarantine and sick leave avoided by following these protocols.

Interventionism and follow-up. Residents and employees were followed up after the first wave of coronavirus. Specifically, a quarterly serology was performed on those who had a positive serology after the first wave. Serological samples were tested with the SARS-CoV-2 IgG antibody test QUANT (Abbott^®^) in order to detect levels of IgG antibodies that bind to the spike protein on the surface of the virus, in serum or plasma. Data interpretation was negative serology when titers were below 50 AU/mL and positive when they were at or above 50 AU/mL [17]. Employees with negative serology were offered a RT according to the “Strategies for the surveillance and control of SARS-CoV-2 in Social and Health Centers” of the Provincial Social and Health Coordination of Albacete (Spain). Regarding the staff screening, this agency recommended to carry out a RT periodically every 10 or 15 days, in order to detect asymptomatic cases of coronavirus.

When a subject was a risk contact of a positive case of coronavirus, the PCRs, quarantines and sick leave avoided were quantified: in the event that the subject presented a positive serology and not a highly clinical suspicious, it was not necessary to perform a PCR or quarantine (or sick leave in the case of an employee). This followed the current recommendations of the Spanish Ministry of Health (Figure 1). On the other hand, if the subject had a negative serology or a high clinical suspicion of coronavirus, then two PCR tests had to be performed (one as soon as possible and another before returning to work after quarantine). In addition, if the subject was an employee, he or she received sick leave (to allow the employee to quarantine).

We define efficiency as the ability to deliver quarterly serial serology (to residents and employees) in order to avoid unnecessary expenditure of material resources (in the event of a coronavirus outbreak). In order to measure efficiency, we compare the economic cost of performing quarterly serological serialization with the savings rendered by not having to perform AIDT, or granting sick leave, to positive IgGs. Our analysis assumes the following diagnostic tests cost from the Castilla-La Mancha Health Service: PCR $41.5/unit, RT $5/unit and serology $5.5/unit. The cost of sick leave was calculated according to data from the Provincial Council of Albacete (Spain).

Statistics. The frequencies of all variables were described. For the quantitative variables, such as age, the mean and standard deviation were calculated. Qualitative variables were expressed as percentages in each category (such as male or female for sex variable). We performed a bivariate and multivariate analysis using binary logistic regression to determine the relationship between age, sex and type of employee (Non-health/Health) and the probability of having a positive IgG serology or PCR result. Odds Ratio (OR) and Adjusted Odds Ratio (aOR) with their respective 95% confidence intervals were estimated. Data were analyzed using SPSS^®^ 22.0 for Windows^®^.

Ethical considerations. The study complies with the Declaration of Helsinki of the World Medical Association on ethical principles for medical research in human beings (version 2008), and the standards of good clinical practice and current legal regulations (Organic Law 3/2018 of 5 December, Protection of Personal Data and guarantee of digital rights and Biomedical Research Law 14/2007). The anonymity of the participants has been maintained and it has been approved by the Drug Research Ethics Committee of Albacete.

## 3. Results

In total, 107 residents and 261 employees were selected. The mean age of the residents was 82.6 years (R 56-99, SD 9.9) and 73.8% were women. Most frequent resident pathology were high blood pressure (67.4%), dementia (59.9%), diabetes (39.6%), anemia (30.0%) and dyslipidemia (28.9%). The mean age of the employees was 51.1 years old (R 23-69, SD 10.3) and 82.4% were women. Job positions were categorized as follows: Nursing assistant 59.2%, cleaning staff 8.0%, nursing 6.5%, doctor 1.5%, kitchen staff 0.4% and others 24.4%.

The results of the diagnostic tests for coronavirus obtained in the first wave are summarized in Table 1: 9.2% of the employees and 13.1% of the residents had a positive PCR. 24.0% of the employees and 51.4% of the residents had a positive IgG serology. Of the employees and residents with positive PCR, 90.9% and 98.1% developed positive IgG serology, respectively. 25% of residents who did not undergo PCR and 56.9% who had negative PCR had subsequent positive serology. Serology was not performed in 2.3% of the employees.

After the first wave, seronegativization (negative serology in people who previously had positive serology) was observed in 7.6% of employees and 1.6% of residents. In September 2020 there was only an employee with a positive PCR, but in October 2020 there was an outbreak in the nursing home: the complementary tests, quarantines and sick leave that were avoided are detailed in Table 2. Among residents, 110 PCR tests were avoided during the October 2020 outbreak, because 55 residents had recent positive serology and they did not have symptoms of high suspicion. During the second wave there were four residents with positive PCR, and none of them had a positive serology at a later time. Among the surviving residents from the first wave, there was a 7.5% death rate until the conclusion of the study.

855 RT were performed on 74% of the employees, but not necessarily every 10–15 days because some of them did not request it. In 24.8% of employees, RT was not indicated, which represents 287 RT avoided (under the assumption that they would have requested them with the same proportion as employees with negative IgG). 1.2% of the employees did not undergo RT because their contract was less than 10 days, and they had a serology or PCR test prior to starting work. Of the RT carried out by screening, only one was positive, which represents a positivity of rapid tests of 0.52% of the employees and 0.12% of the total rapid tests carried out.

The financial report is detailed in Table 3. Estimated savings totaled $14,753.5. Savings were higher for employees ($10,456) than for residents ($4297.5). The greatest savings were the days of sick leave avoided for nursing assistants ($8700) and the greatest expense was the performance of RT in employees ($4275). The lowest cost was performing serology tests on residents ($599.5).

Table 4 and Table 5 show, both in the bivariate and multivariate analyses, a higher probability of having a positive serology or PCR for COVID-19 in healthcare employees compared to non-healthcare employees: with an aOR of 5.47 (95% CI: 2.22–13.51) in the case of positive IgG, and with an aOR of 3.79 (95% CI: 1.07–13.42) in the case of positive PCR.

## 4. Discussion

The serological serialization of coronavirus on a quarterly basis, in residents and employees of our nursing home, has proven to be efficient in avoiding unnecessary expenditure of AIDT during a coronavirus outbreak, as well as avoiding quarantines and sick leave of participants with positive IgG. 

The WHO issued a guide for nursing homes, indicating that they must take special precautions to protect their residents, employees and visitors [18]. These documents recommend actions to minimize the effect of prevention on the mental health of residents and employees, such as avoiding unnecessary quarantines. It should be taken into account that isolation and social distancing have had and are having a great impact on this population, due to the changes in their lives. This is added to the psychosocial effects directly related to the pandemic (anxiety, depression, sleep disorders, and so on) [19]. Following the instructions of the Spanish Ministry, in our nursing home 15 quarantines have been avoided in workers and 59 in residents. This also avoids the employees need to take sick leave, which in turn helps the nursing home to keep sufficient levels of healthy staff to meet residents’ needs. Staff shortages are known as one of the side effects on nursing homes during the first wave [20].

During the period in which the study was carried out, there was no excessive alarm about the possibility that the virus had the capacity for reinfection, since the data on reinfections worldwide were infrequent and controversial [21]. Thus, it seemed reasonable that doctors did not perform PCR on patients who had positive IgG serology in the previous three months. In recent months, these criteria have changed due to the appearance and spread of new strains that have increased the probability of reinfections, and due to the high percentage of vaccinated in Spain (greater than 90%, with so much population vaccinated, it is expected that they will have positive serology.). So, last recommendations of Resolution of 18 May 2022 of the Ministry of Health of Castilla-La Mancha (Spain) are: employees of health and socio-health centers are considered as suspected cases of COVID-19 if they present compatible symptoms, even if they have a positive IgG serology drawn in the last three months. In addition, they recommend performing 2 weekly SARS-CoV-2 Rapid Antigen Test Nasal in unvaccinated employees (even if they do not have symptoms or have been close contacts) and PCR 3–5 days after close contact for COVID-19 (Figure 2). However, in other countries with lower vaccination rates, such as China, or with less access to AIDT, it is interesting to assess serial serology in avoiding unnecessary expenditure of AIDT.

The first published COVID-19 outbreak in a long-term care facility included 130 residents and 170 employees: after the first detected case, 77.7% of residents and 29.4% of healthcare employees were confirmed by COVID-19 [19]. In our center there were fewer cases of COVID-19 among residents (51.4%) but similar in health employees (30.5%). The percentage of positive serologies was lower (10.6%) in non-health employees (cleaning, administration, maintenance, caretakers, laundry, etc.), probably because they had less direct contact with the airways of the residents. About our employees, the mean age of the healthcare employees with positive IgG serology was 51.0 years, this was less than in a meta-analysis (38.37 years) [22]. It was expected that healthcare employees would have a higher positive PCR or serology (Table 4 and Table 5) than non-healthcare employees because they are in more direct contact with COVID-19 patients [7]. It should be noted that, among residents with negative and non-performed PCR in the first wave, 25% and 56.9% (respectively) had subsequent positive serology. This suggests that they were asymptomatic cases of COVID-19 during the first wave, which supports the idea of serology screening in institutionalized patients in order to know their immune status. This helps avoiding unnecessary PCR tests during an outbreak, which are costly and invasive for the resident. In our sample, seronegativization (undetectable IgG antibodies in people who were previously seropositive) after the first wave has been 0.8% in residents and 7.4% in employees, compared to the range 7.1–14.4% at the national level [7]. This low percentage of seronegativity in residents stands out, so it would be interesting to analyze (in a larger study) whether or not age or institutionalization are protective factors for seronegativity. Available data from Italy, China and the UK show that the most common comorbidities seen in COVID-19 patients are hypertension (63.1–74.7%), dementia (56.6%) and diabetes (22.0–30.5%) [23,24]. In our study, we obtained similar results: high blood pressure (67.4%), dementia (59.9%) and diabetes (39.6%).

We note the reported economic savings (almost $15,000) is an underestimation given savings from human resources (transportation and laboratory personnel) were not included in the cost of diagnostic tests. The greatest expense was the performance of rapid tests ($4275). Of the 855 performed, only one rapid test was positive, which after completing the study with serology and PCR, was concluded to be a false positive. It is precisely those cases that make screening asymptomatic social and health workers questionable; in fact, the WHO does not recommend them [16] and the Provincial Social and Health Coordination of Albacete stopped recommending them on 19 March 2021. Nonetheless, at least in 2020, it was correct to indicate that RT should not be performed on workers with recent positive serologies (in the last 3 months) since it would duplicate already known information.

Our study has some limitations, such as the fact that it was only carried out in one nursing home. It would be interesting to include results from other centers in future research. Another important limitation at the beginning of the first wave was the lack of PCR confirmation from all residents, because they were only performed on hospital patients. Even at a later time, when it was possible to perform PCR in nursing homes, Albacete suffered a shortage of PCR reagents [25]. As a consequence, doctors were very strict and only considered “probable” cases, i.e., when compelling symptoms and there was a clear and recent risk contact. For this reason, it would be justified that 25% of the residents who did not undergo PCR had subsequent positive serology, because they were asymptomatic cases or with very mild symptoms. Another limitation could be the extrapolation of our results to other countries due to differences in nursing home models.

Nursing homes need to continue with the support of socio-health institutions and professionals, and this need for support will not end when the pandemic is resolved [26]. In this scenario, knowledge about the cost-effectiveness of complementary tests for the diagnosis and monitoring of COVID-19 is essential to help make reasonable use of the material resources available. We conclude that surveillance for positive COVID-19 serology among long-term care staff and residents may be a cost-effective strategy during a pandemic. In addition, we have detected a low utility of RT, a higher probability of having a positive PCR or serology for COVID-19 in healthcare employees compared to non-healthcare employees and a high number of asymptomatic cases of COVID-19 during the first wave. Our work emphasizes the need to know and apply strategies for early detection, surveillance, and control of COVID-19 for future outbreaks, because the institutionalized elderly is the population at highest risk.

## Figures and Tables

**Figure 1 geriatrics-07-00078-f001:**
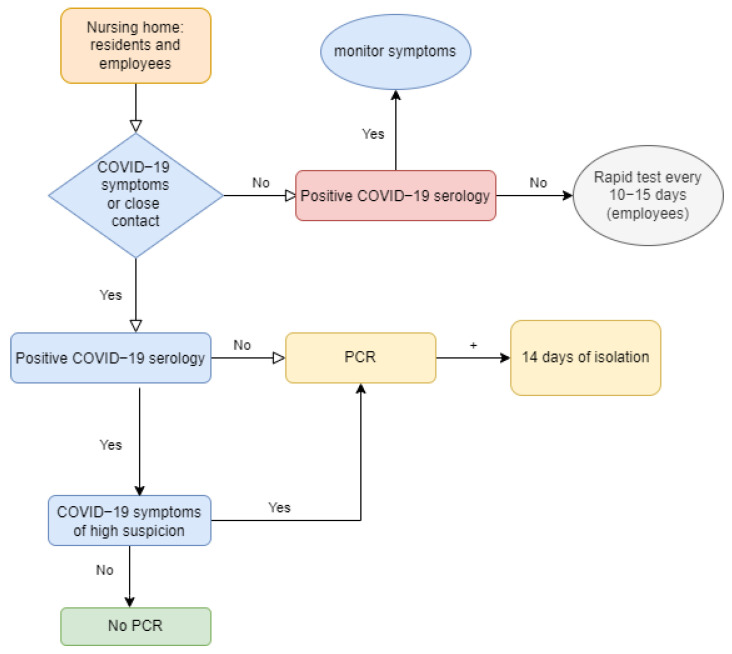
Suspected cases of COVID-19, recommendations of the Spanish Ministry of Health in 2020.

**Figure 2 geriatrics-07-00078-f002:**
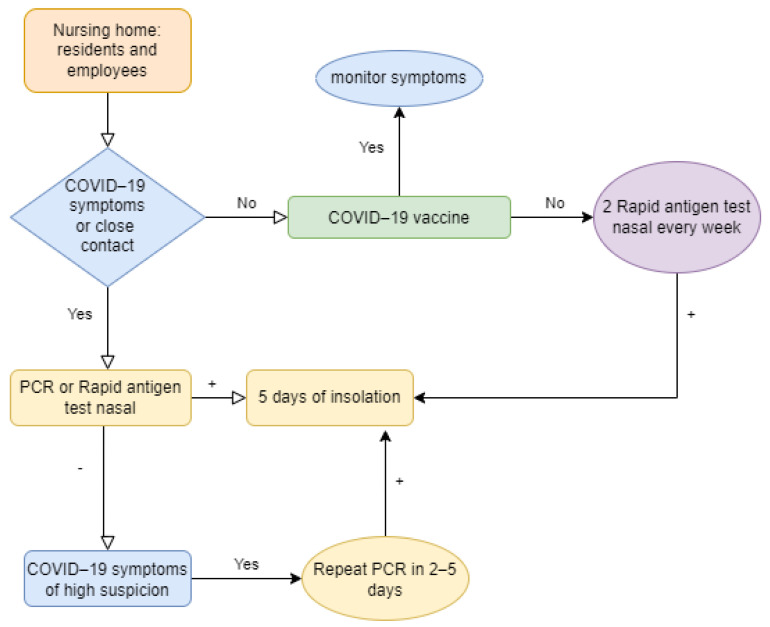
Suspected cases of COVID-19, recommendations of the Spanish Ministry of Health in 2022.

**Table 1 geriatrics-07-00078-t001:** Diagnostic tests for coronavirus in the first wave.

Positive Serology IgG ^2^ after First Wave	PCR ^1^ in First Wave	
51.4% (total)		Residents
98.1%	-Positive: 13.1%	
56.9%	-Negative: 60.7%	
25.0%	-Not performed: 26.2%	
24.0% (total)		Employees
90.9%	-Positive: 9.2%	
8.6%	-Negative: 80.2%	
82.1%	-Not performed: 10.6%	

^1^ Polymerase Chain Reaction; ^2^ Inmunoglobulin G.

**Table 2 geriatrics-07-00078-t002:** Avoided resources in the second wave.

	Employees	Residents
PCR ^1^ avoided:		
-September 2020	4 (2 workers)	
-November 2020	14 (7 workers)	
Quarantines avoided:		
September 2020	9 (5 nursing assistants, 3 nursing and 1 cleaning staff)	4
October 2020	6 (5 nursing assistants and 1 nursing)	55

^1^ Polymerase Chain Reaction.

**Table 3 geriatrics-07-00078-t003:** Financial report.

	Resource	N	Cost Per Unit	Total
Residents saving	PCR ^1^ avoided	118	$41.5	$4.897
Residents cost	COVID-19 serology	109	$5.5	$599.5
Residents balance				$4297.5
Employees saving	PCR ^1^ avoided	18	$42	$756
	Rapid tests avoided	287	$5	$1435
	Days of work avoided:			
	Nursing	40	$121.5	$4860
	Nursing assistants	100	$87	$8700
	Cleaning staff	10	$75	$750
Employees cost	Serology	295	$6	$1770
	Rapid test	855	$5	$4275
Employees balance				$10,456
Total savings				$14,753.5

^1^ Polymerase Chain Reaction.

**Table 4 geriatrics-07-00078-t004:** Relationship between positive PCR and the professional profile.

Positive PCR ^1^				
aOR 95% CI	OR 95% CI	Yes, n (%)	No, n (%)	
0.99 (0.95–1.03)	0.99 (0.95–1.03)	49.9 (11.3)	51.2 (10.2)	Age (Mean SD)
				Sex:
1	1	4 (9.3)	39 (90.7)	-Men
0.75 (0.23–2.45)	10.96 (0.32–3.08)	20 (9.3)	196 (90.7)	Women
1	1	3 (3.5)	82 (96.5)	Employees:
				-Non-healthcare
				-Healthcare
3.79 (1.07–13.42)	3.70 (1.07–12.78)	21 (11.9)	155 (88.1)	

^1^: Polymerase Chain Reaction.

**Table 5 geriatrics-07-00078-t005:** Relationship between positive IgG serology and the professional profile.

Positive IgG ^1^ Serology				
aOR 95%CI	OR 95%CI	Yes, n (%)	No, n (%)	
0.99 (0.95–1.03)	0.99 (0.95–1.03)	51.0 (10.3)	51.1 (10.3)	Age (Mean SD)
				Sex:
1	1	8 (18.6)	35 (81.4)	
1.16 (0.46–2.92)	1.46 (0.64–3.34)	54 (25.0)	162 (75.0)	Men
1	1	9 (10.6)	76 (89.3)	Women
5.47 (2.22–13.51)	3.71 (1.73–7.94)	54 (30.5)	123 (69.5)	Employees:
				-Non-healthcare
				-Healthcare

^1^: Inmunoglobulin G.

## Data Availability

The principal investigator, Sergio Salmerón, has all the study data.

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
