# Peer review of "Efficiency of Diagnostic Test for SARS-CoV-2 in a Nursing Home"

_geriatrics, 2022, doi:10.3390/geriatrics7040078_

Round 1

Reviewer 1 Report

First of all, I would like to congratulate the authors for their work. 

I would suggest some improvement areas of this article:

Regarding the introduction:

- I found the introduction too enlarged. I suggest to try to make it more concise.

- I found the fourth paragraph (that start at line number 53) confusing. I suggest to try clarify which test and when was recommended by Spanish authorities. Or at least refer to figure 1 where the information is clearer. 

- At line number 54 the word "heath" should be changed to "health".

- Probably paragraph five (started at line number 71) should be moved to the "Material and Methods" section. 

Regarding the results: 

- Paragraph two (started at line number 169) would be better understood with a flow-chart figure. 

- At line number 183: 110 PCR test were avoided because 55 residents had recent positive serology. It means that none of the 55 residents had COVID-19 symptoms or high suspicion? (according to figure 1 it was recommended to test with PCR those people with symptoms or high suspicion). 

From my point of view this article provides interesting information and can be accepted after minor structure and content revision. 

Author Response

Note: Answers in black

First of all, I would like to congratulate the authors for their work.

Thanks a lot.

I would suggest some improvement areas of this article:

Regarding the introduction:

- I found the introduction too enlarged. I suggest to try to make it more concise.

The authors have reduced the introduction: previously, the introduction had 736 words, now 656 words.

- I found the fourth paragraph (that start at line number 53) confusing. I suggest to try clarify which test and when was recommended by Spanish authorities. Or at least refer to figure 1 where the information is clearer.

This paragraph has been corrected for better understanding, adding dates and the reference of figure 1 has been added for clearer information.

- At line number 54 the word "heath" should be changed to "health".

We have corrected it.

- Probably paragraph five (started at line number 71) should be moved to the "Material and Methods" section.

The reviewer is right, this paragraph has been integrated into material and methods to avoid duplicating information.

Regarding the results:

- Paragraph two (started at line number 169) would be better understood with a flow-chart figure.

Paragraph two and Table 1 have been updated for better understanding. We have tried to make a flow-chart but we believe that a table is more accessible.

- At line number 183: 110 PCR test were avoided because 55 residents had recent positive serology. It means that none of the 55 residents had COVID-19 symptoms or high suspicion? (according to figure 1 it was recommended to test with PCR those people with symptoms or high suspicion).

Of course, we have uptade this line “Among residents, 110 PCR tests were avoided during the October 2020 outbreak, because 55 residents had recent positive serology and they did not have symptoms of high suspicion”

From my point of view this article provides interesting information and can be accepted after minor structure and content revision.

Thanks a lot.

Reviewer 2 Report

The study seems interesting, although it suffers from serious flaws.

First of all, it has no conclusions, and the proposals it makes after the limitations are not derived from the study.

As for the study, it is based on simple averages, which do not provide any knowledge. more elaborate statistics are needed, with odd ratio, regressions, inferences, something that allows valid conclusions to be drawn.

It should state the objectives well, specifically and clearly.

The description of the sample and methodology is vague and imprecise.

It really needs a deep modification.

Author Response

Note: Answers in black

The study seems interesting, although it suffers from serious flaws.

We have update the paper: we have had the help of a statistician university professor (Antonio Hernández), expanding the methodology, objectives, results and discussion.

First of all, it has no conclusions, and the proposals it makes after the limitations are not derived from the study.

Conclusions have been added in the last paragraph of the discussion.

The proposals that we make after the limitations has been deleted.

As for the study, it is based on simple averages, which do not provide any knowledge. more elaborate statistics are needed, with odd ratio, regressions, inferences, something that allows valid conclusions to be drawn.

This is very interesting. Our originally study was a descriptive study where the objective is to analyze whether serological serialization is cost-effective, making an economic memory based on the balances of spending and savings. But the reviewer is right that we can do more statistics and we have requested the collaboration of a statistician (Antonio Hernández, university professor). So we have included a secondary objective: To evaluate, in employees, if the COVID-19 infection (PCR or positive serology) is related to age, sex or job position. Their results are in table 4 and 5, also we have included this new data in the discussion.

It should state the objectives well, specifically and clearly.

We have updated the objectives in the last paragraph of the introduction.

The description of the sample and methodology is vague and imprecise.

Sample information has been updated. For de employeers we selected 261, we think that is a good sample because we used GRANMO for we estimated this sample (https://www.imim.es/ofertadeserveis/software-public/granmo/): A sample size of 241 subjects randomly selected will suffice to estimate with a 95% confidence and a precission +/- 4 percent units, a population percentage considered to be around 10%. It has been anticipated a replacement rate of 10%.

We added in methodology-statistics: “We performed a bivariate and multivariate analysis using binary logistic regression to determine the relationship between age, sex and type of employee (Non-health/Health) and the probability of having a positive IgG or PCR result. Odds Ratio (OR) and Adjusted Odds Ratio (aOR) with their respective 95% confidence intervals were estimated.”

It really needs a deep modification.

We thank the reviewer for their input as it has improved the publication.

Reviewer 3 Report

I appreciate the invite to review this research article for Geriatrics MDPI Journal. This cost-efficiency retrospective observational study is of very interest to elaborate the best strategy in the COVID-19 outbreaks, even if the sample size is limited as well as the involvement of other nursing homes. These limits are well recognized from the authors. In addition, I consider that more data and analyses can implemented the study. For example, the results of quantitative PCR assays for SARS-CoV-2 would have been useful in this type of study to evaluate the real risk of viral dissemination by people without symptoms and with close contact and a positive serology IgG.

Please you to specify the rapid test mentioned in the abstract in the line 15 as well as in the main body in the line 46. Antigen or antibody? It should be specified in the first mention.

At line 58-63 it is low clear the significance of the sentence. People with COVID-19 symptoms with previous SARS-CoV-2 infection history don’t be considered as suspected cases except for those with symptoms… In the same way for the lines 63-66, it is low clear about the performance of molecular or antigen direct assays.

In the line 152, they are missing the specific statistical tests used in the data analysis with SPSS software, if they are.

In the line 237-239 it is unclear that the Rapid Antigen Tests are reserved to people without vaccine and symptoms or close contact for COVID-19. The figure explains the concepts but the text must be improved for the same objective.

Author Response

Note: Answers in black

I appreciate the invite to review this research article for Geriatrics MDPI Journal. This cost-efficiency retrospective observational study is of very interest to elaborate the best strategy in the COVID-19 outbreaks, even if the sample size is limited as well as the involvement of other nursing homes. These limits are well recognized from the authors. In addition, I consider that more data and analyses can implemented the study. For example, the results of quantitative PCR assays for SARS-CoV-2 would have been useful in this type of study to evaluate the real risk of viral dissemination by people without symptoms and with close contact and a positive serology IgG.

The reviewer is right, we have added more data and analysis in the study. We have requested the collaboration of a statistician (Antonio Hernández, university professor). Your example is very interesting, but we can do it because we did not have the data of people without symptoms and positive PCR. So we have included a secondary objective: To evaluate, in employees, if the COVID-19 infection (PCR or positive serology) is related to age, sex or job position. Results are in Table 4 and 5. Also, we have included more data in Table 1.

Please you to specify the rapid test mentioned in the abstract in the line 15 as well as in the main body in the line 46. Antigen or antibody? It should be specified in the first mention.

We have specify it. It was an antibody rapid test.

At line 58-63 it is low clear the significance of the sentence. People with COVID-19 symptoms with previous SARS-CoV-2 infection history don’t be considered as suspected cases except for those with symptoms… In the same way for the lines 63-66, it is low clear about the performance of molecular or antigen direct assays.

That paragraph has been rewritten for better understanding: “people with symptoms compatible with COVID-19 who have already had an infection confirmed by active coronavirus infection detection test (AIDT: PCR or RT of antigens) or IgG serology of positive SARS-CoV-2 in the previous 90 days, would not be considered suspected cases again, except if they exhibited COVID-19 symptoms of high suspicion”. And in the line 63-66 we change IgG for “IgG antibody serology”.

In the line 152, they are missing the specific statistical tests used in the data analysis with SPSS software, if they are.

This has been included in the methodology (Statistics).

In the line 237-239 it is unclear that the Rapid Antigen Tests are reserved to people without vaccine and symptoms or close contact for COVID-19. The figure explains the concepts but the text must be improved for the same objective.

The reviewer is right, we have clarified those lines: “they recommend performing 2 weekly SARS-CoV-2 Rapid Antigen Test Nasal in unvaccinated employees (even if they do not have symptoms or have been close contacts)”
